# Stimuli-Responsive Thiomorpholine Oxide-Derived Polymers with Tailored Hydrophilicity and Hemocompatible Properties

**DOI:** 10.3390/molecules27134233

**Published:** 2022-06-30

**Authors:** Laura Vasilica Arsenie, Franziska Hausig, Carolin Kellner, Johannes C. Brendel, Patrick Lacroix-Desmazes, Vincent Ladmiral, Sylvain Catrouillet

**Affiliations:** 1ICGM, University of Montpellier, CNRS, ENSCM, Montpellier, France; laura-vasilica.arsenie@umontpellier.fr (L.V.A.); patrick.lacroix-desmazes@enscm.fr (P.L.-D.); 2Laboratory of Organic and Macromolecular Chemistry (IOMC), Friedrich Schiller University Jena, 07743 Jena, Germany; franziska.hausig@uni-jena.de (F.H.); carolin.kellner@uni-jena.de (C.K.); johannes.brendel@uni-jena.de (J.C.B.); 3Jena Center for Soft Matter (JCSM), Friedrich Schiller University Jena, Philosophenweg 7, 07743 Jena, Germany

**Keywords:** pH-responsive polymers, temperature responsive polymers, thiomorpholine oxide, RAFT, cytotoxicity

## Abstract

Thermo-responsive hydrophilic polymers, including those showing tuneable lower critical solution temperature (LCST), represent a continuous subject of exploration for a variety of applications, but particularly in nanomedicine. Since biological pH changes can inform the organism about the presence of disequilibrium or diseases, the development of dual LCST/pH-responsive hydrophilic polymers with biological potential is an attractive subject in polymer science. Here, we present a novel polymer featuring LCST/pH double responsiveness. The monomer ethylthiomorpholine oxide methacrylate (*THOXMA*) can be polymerised via the RAFT process to obtain well-defined polymers. Copolymers with hydroxyethyl methacrylate (*HEMA*) were prepared, which allowed the tuning of the LCST behaviour of the polymers. Both, the LCST behaviour and pH responsiveness of hydrophilic *PTHOXMA* were tested by following the evolution of particle size by dynamic light scattering (DLS). In weak and strong alkaline conditions, cloud points ranged between 40–60 °C, while in acidic medium no LCST was found due to the protonation of the amine of the *THOX* moieties. Additional cytotoxicity assays confirmed a high biocompatibility of *PTHOXMA* and haemolysis and aggregation assays proved that the thiomorpholine oxide-derived polymers did not cause aggregation or lysis of red blood cells. These preliminary results bode well for the use of *PTHOXMA* as smart material in biological applications.

## 1. Introduction

Smart polymers which react upon external stimuli are attractive materials in biomedical applications and promise for example the selective release of encapsulated drugs only under specific conditions or in targeted tissue or cells [1]. LCST behaviour is one unique feature polymers may demonstrate, which can be exploited in this regard [2]. Lower critical solution temperature (LCST) in water is an important property for polymers destined to biological applications. Below the LCST, the polymer is soluble due to the H-bonds formed with water molecules [3]. Above the LCST, these H-bonds are disrupted and polymer chain aggregation takes place as a result of non-covalent interactions between hydrophobic moieties of polymer [4]. In addition, above the cloud point, the entropy is a dominating factor which results in the release of water molecules and turns into the collapse and coagulation of the polymer chains [4].

Biocompatible homopolymers exhibiting an LCST in the physiological relevant range (30–40 °C), [5] such as POEGMA (poly(oligoethylene glycol methacrylate)), polyoxazolines or PNIPAM (poly(*N*-isopropyl acrylamide)) have been thoroughly investigated and tested in biomedical applications [6,7,8]. In the case of OEGMA, an increase in EG units decreases the LCST. For oxazolines, long alkyl chains led to low LCST values. In addition, HEMA is a co-monomer that could be used to tune the LCST of responsive polymers. For example, Zhang et al. reported that an increase in the HEMA molar fraction from 1% to 7% in poly(*N*-isopropyl acrylamide-*co*-acrylamide–*co*-hydroxyethyl methacrylate) P(NIPAM-*co*-AM-*co*-HEMA) terpolymers led to an LCST decrease from 68.2 °C to 44.7 °C [9,10]. Similar observations were reported by Kasprow and collaborators for poly(hydroxyethyl methacrylate-*co-*oligoethylene glycol methacrylate) P(HEMA-*co*-OEGMA) copolymers [11]. Their research demonstrated that HEMA decreased the LCST of copolymers (from 61.5 °C to 21.5 °C) via intermolecular H-bonds, while hydrophilic OEGMA controlled the hydrophilic/hydrophobic balance and increased the LCST. Poly(2-(*N*-(dimethylamino) ethyl methacrylate) (PDMAEMA) is a biocompatible homopolymer which shows dual pH and thermo-responsivity, which is advantageous in tailoring the LCST [8].

Another important parameter in the development of stimuli-responsive polymers for biomedical applications is the pH. The physiological pH value of blood (for human or animal cells) is 7.4. Changes in this physiological pH are often symptomatic of the presence of infections or cancer [12]. Morpholine-derived polymers contain tertiary amino groups which are protonated in acidic pH (pK_a_ of morpholine = 8.3) [13,14]. This property can be used to tailor the behaviour of the polymer in different biological environments [13,14]. Moreover, poly(*N*-acryloylmorpholine) (PNAM) and copolymers of NAM present a high water solubility and biocompatibility and are thus particularly interesting for biological applications [15]. Furthermore, pH-responsive materials with pK_a_ values ranging between 5 and 7.4 are further interesting candidates for entering the cells and facilitating a pH-induced endosomal escape [13,16].

Previous studies showed that poly(morpholine ethyl methacrylate) (PMEMA) possesses a pK_a_ of 4.9 and displays molar mass-dependent LCST at pH 7 and higher [17,18]. PMEMA and PTHOXMA are interesting pH-responsive platforms since their pK_a_ is below 7.4 and they thus remain neutral at physiological conditions [8]. This property is important since these polymers are possible alternatives to cationic polymers which are currently used in the field of cellular transfection. Indeed, cationic polymers are intrinsically toxic and they have no stealth features since they strongly interact with any cell surface [19].

Replacing morpholine motif with thiomorpholine could result in new polymer platforms with huge potential as responsive materials. However, the thiomorpholine-containing polymers show a relatively low water solubility [20]. Sobotta et al. reported that the oxidation of the sulphur atom of thiomorpholine in poly[(*N*-acryloylmorpholine)-*b*-(*N*-acryloylthiomorpholine)] resulted into poly[(*N*-acryloylmorpholine)-*b*-(*N*-acryloylthiomorpholine oxide)] with high water solubility and no cytotoxicity [20].

To the best of our knowledge, few examples of thiomorpholine oxide-containing polymers have been reported so far, and the dual pH/temperature sensitivities of these polymers have never been explored. In this work, a new class of polymers containing thiomorpholine oxide units, named poly(ethylthiomorpholine oxide methacrylate) *PTHOXMA,* was synthesised by reversible addition−fragmentation chain-transfer (RAFT) polymerisation. This polymer presents protonable tertiary amine functions which are sensitive to pH and sulfoxide groups which confer a strong hydrophilic character. To gain deeper insight into the double LCST/pH responsivity of this new polymer class, the present study also explores how copolymerisation of *THOXMA* with *HEMA* can be used to modulate the LCST of the resulting copolymers. HEMA was chosen since it is a common co-monomer used to modify the LCST behaviour and because it could easily statistically copolymerise with *THOXMA*. The double LCST/pH responsivity was monitored by DLS at different pHs (4, 7.4 and 10). Cytotoxicity, haemolysis tests and cell aggregation rate assays were performed to evaluate the biocompatibility of *PTHOXMA*.

## 2. Experimental Section

### 2.1. Materials

Methacryloyl chloride (97% purity) was purchased from Fluka (France) and distilled at 50 °C, 400 mbar before use. Hydrochloric acid solution (37 wt%), sodium bicarbonate (NaHCO_3_) and magnesium sulphate (MgSO_4_) were obtained from Fluka (France). 2-bromoethanol (95% purity) was purchased from Alfa Aesar. Thiomorpholine (98% purity) was acquired from Fluorochem and hydrogen peroxide (H_2_O_2_) solution (30 wt%) from Carlo Erba. Hydroxyethylmethacrylate (95% purity, HEMA), 4-dimethylaminopyridine (99% purity, DMAP), triethylamine (99% purity, TEA), 2-cyano-2-propyl benzodithioate (CPDB), 2,2′-azobis(2-methylpropionitrile) (AIBN) (recrystallised from methanol at 65 °C before use in the polymer synthesis), potassium carbonate (K_2_CO_3_), disodiumphosphate basic dodecahydrate (Na_2_HPO_4_·12H_2_O) and deuterated solvents (CDCl_3_ and DMSO-d_6_) were provided by Sigma Aldrich. Sodium chloride (NaCl) and citric acid monohydrate (C_6_H_8_O_7_·H_2_O) were obtained from VWR Chemical. Dimethyl sulfoxide (98% purity, DMSO) was obtained from Acros Organics. Dry solvents (dichloromethane, 95% purity, CH_2_Cl_2_, and acetonitrile, 99% purity, CH_3_CN) were dried on a solvent purification system PureSolv Micro (Sigma Aldrich). The dialysis membranes used for purification of polymers (Spectra/Por 7 Preteated RC Dialysis Tubing, MWCO = 1 kDa, diameter 24 mm, 4.6 mL/cm) were bought from Krackeler Scientific, USA. In total, 2 mM L-glutamine 100 U/mL penicillin and 100 μg/mL streptomycin solutions were achieved from Biochrom. In total, 10% fetal calf serum was bought from FCS, Capricorn Scientific (Ebsdorfergrund, Germany). The PrestoBlue solution was obtained from Thermo Fisher, Germany. Sheep blood was provided by the Institute for Experimental Animal Science and Animal Welfare, Jena University Hospital. Branched poly(ethylene imine) (bPEI) solution was purchased from Polysciences Inc. (Warrington, PA, USA).

### 2.2. Instrumentation

^1^H-NMR spectra were recorded on NMR Bruker Avance 400-MHz or III HD-400 MHz spectrometers using CDCl_3_ or DMSO-*d*_6_ as deuterated solvent. The chemical shifts of protons were relative to tetramethylsilane (TMS) at *δ* = 0.

Size exclusion chromatography (SEC) data were obtained in DMF containing 0.1 wt% LiCl, with a flow rate of 0.8 mL/min at 40 °C. Samples were filtered using TE36 Whatman PTFE-supported membrane filter paper (0.45 µm, 47 mm diameter) before the injection. The data were calibrated using poly(methyl methacrylate) (PMMA) narrow standards.

Fourier transform infrared spectroscopy (FTIR) analysis was achieved with a Perkin Elmer Spectrum 100 spectrometer. The spectral data were acquired in the 3500–500 cm^−1^ range.

Dynamic light scattering (DLS) analyses were performed on a Malvern Zetasizer NanoZS instrument at a detection angle of 173° (back scattering), in the 10–60 °C temperature range.

Cytotoxicity and hemolysis assays were performed using a Tecan plate reader (Tecan, Männedorf, Switzerland). The fluorescence measurements used to determine the cell viability were assessed using an Infinite M200 PRO microplate reader from Tecan, Germany.

### 2.3. Synthesis of 2-Bromoethyl Methacrylate

2-bromoethyl methacrylate was prepared according to a published procedure [21]. Briefly, to a solution of 4-(dimethylamino)pyridine (DMAP) (103.4 mg, 0.846 mmol, 0.05 eq.) in CH_2_Cl_2_ (100 mL), 2-bromoethanol (1.2 mL, 17 mmol, 1 eq.) and triethylamine (TEA) (4.7 mL, 34 mmol, 2 eq.) were added under continuous stirring. Then, methacryloyl chloride (1.66 mL, 17 mmol, 1 eq.) was added dropwise in an ice bath and under inert (N_2_) atmosphere. The reaction mixture was kept at room temperature and under inert atmosphere overnight. The resulting mixture was washed twice with a saturated NaHCO_3_ aqueous solution (2 × 100 mL) and then with distilled water (100 mL). The organic layer was collected, dried with MgSO_4_, filtered and concentrated under vacuum to give a brown oil (3 g, yield: 89%).

### 2.4. Synthesis of Ethylthiomorpholine Methacrylate (THMA)

Anhydrous K_2_CO_3_ (2.14 g, 15.5 mmol, 1 eq.) was dissolved in dry acetonitrile (100 mL) under stirring for 30 min. Then, thiomorpholine (1.56 mL, 15.5 mmol, 1 eq.) was added and the reaction mixture was stirred for another 30 min. Subsequently, freshly prepared 2-bromoethyl methacrylate (3 g, 15.5 mmol, 1 eq.) was added and the reaction mixture was kept under inert atmosphere for 6 days at 40 °C using an oil bath. The solvent was then evaporated using a rotary evaporator. Then, dichloromethane was added to the resulting oil and the mixture was washed 6 times with distilled water (6 × 100 mL). The organic phase was collected, dried with MgSO_4_, filtered and concentrated under vacuum to give a viscous liquid (*THMA*, 2.77 g, yield: 83%). ^1^H NMR for *ethylthiomorpholine methacrylate* (400 MHz, CDCl_3_, Appendix A) *δ* (ppm) = 6.01 (d, C**H**_**2**_, noted as a); 5.58 (d, C**H**_**2**_, noted as a’); 1.87 (s, C**H**_3_, noted as b); 4.18 (t, OC**H**_2_CH_2_N, noted as c); 2.66 (t, OCH_2_C****H**_**2**_**N, noted as d); 2.72 (t, NC**H**_2_CH_2_S, cyclic thiomorpholine, noted as e); 2.58 (t, NCH**_2_**C**H****_2_**S, cyclic thiomorpholine, noted as f).

### 2.5. Synthesis of Ethylthiomorpholine Oxyde Methacrylate (THOXMA)

2-thiomorpholine ethyl methacrylate (2.77 g, 12.8 mmol, 1 eq.) was put in a single neck round-bottom flask sealed with a rubber stopper. The flask containing the product was kept in an ice bath and purged with N_2_ for 15 min, then a 30 wt% hydrogen peroxide solution (1.44 mL, 14.1 mmol, 1.1 eq.) was slowly added and the reaction mixture was stirred for 24 h. The reaction was then diluted with 50 mL of deionised water. The aqueous solution was washed 3 times with 100 mL of dichloromethane. The organic phase was collected, dried over MgSO_4_ and then dried under vacuum, resulting in an orange liquid (*THOXMA*, 2.37 g, yield: 80%).^1^H NMR for *ethylthiomorpholine oxide methacrylate* (400 MHz, CDCl_3_, Appendix A) *δ* (ppm) = 6.01 (d, C**H**_**2**_, noted as a); 5.58 (d, C**H**_**2**_, noted as a’); 1.87 (s, C**H**_3_, noted as b); 4.18 (t, OC**H**_2_CH_2_N, noted as c); 3.2 (t, OCH_2_C**H_**2**_**N, noted as d); 2.85 (t, NC**H**_2_CH_2_S, cyclic thiomorpholine, noted as e); 2.93 (t, NCH**_2_**C**H_2_**S, cyclic thiomorpholine, noted as f).

### 2.6. Synthesis of Poly(ehylthiomorpholine oxide methacrylate) P(THOXMA)_100_ Homopolymer and Statistical Copolymers Poly(ethylthiomorpholine oxide methacrylate-stat-hydroxyethylmethacrylate) P(THOXMA_n_-stat-HEMA_m_) by RAFT

The general procedure used to prepare *P(THOXMA)_100_* homopolymer was as follows. A 10 mL ampoule was charged with *THOXMA* (2 g, 8.65 mmol, 100 eq.), CPDB (19 mg, 0.085 mmol, 1 eq.) and AIBN (0.35 mg, 0.002 mmol, 0.25 eq.) and then dissolved in 1.5 mL DMSO. The mixture was degassed via three freeze pump thaw cycles. The ampoule was then filled with nitrogen and immersed in an oil bath at 75 °C. Every hour, an aliquot was taken and analysed by ^1^H NMR and SEC. After 6 h (conversion 99.3%), the reaction was stopped by exposure to air. The mixture was dialysed against water (with a 1 kDa MWCO membrane) for 2 days, followed by lyophilisation for 1 day. The resulting pink powder (yield: 80%) was analysed by ^1^H NMR and SEC. ^1^H NMR (400 MHz, DMSO-d_6_, Figure 2) *δ* (ppm) = 0.94–1.5 (m, C**H**_3_, polymer backbone, noted as a); 1.8 (br s, C**H**_**2**_, polymer backbone, noted as b); 2.63 (br s, C**H**_**2**_-S, noted as f); 2.88 (br s, C**H**_**2**_-N, noted as d); 2.97 (br s, C**H**_**2**_-N in the thiomorpholine cycle, noted as e); 3.99 (br s, C**H**_**2**_-O, noted as c).

Following the previous protocol, a range of statistical copolymers of 2-thiomorpholine oxide ethyl methacrylate (*THOXMA*) and 2-hydroxyethyl methacrylate (HEMA) *P(THOXMA_n_-stat-HEMA_m_)* were prepared. For example, *P(THOXMA_50_-stat-HEMA_50_)* was prepared as follows. *THOXMA* (2 g, 8.65 mmol, 50 eq.), *HEMA* (1.12 g, 8.65 mmol, 50 eq.), CPDB (38 mg, 1 eq.), AIBN (7 mg, 0.25 eq.) and 1.5 mL of DMSO were mixed in a 10 mL ampoule. The mixture was degassed via three freeze pump thaw cycles, the ampoule was filled with nitrogen and then immersed in an oil bath at 75 °C. An aliquot was taken every hour for ^1^H-NMR and SEC-HPLC analyses. After 4 h (conversion above 99% for both monomers), the reaction was quenched by the exposure of the mixture to air. At the end, the mixture was dialysed against water (with a 1 kDa MWCO membrane) for 2 days, followed by lyophilisation for 1 day. The pink polymer (yield: 78%) was analysed by ^1^H NMR and SEC-HPLC. ^1^H NMR (400 MHz, DMSO-d_6_, Appendix A) *δ* (ppm) = 0.97–1.6, (m, C**H**_3_, polymer backbone, noted as g); 1.77 (br s, C**H**_**2**_, polymer backbone, noted as h**);** 2.6–2.8 (br m, C**H**_**2**_-N, noted as c and e); 2.97 (CH_2_C**H**_**2**_-SO, noted as f); 3.57 (br s, C**H**_**2**_-O, HEMA, noted as b); 3.86 (br s, C**H**_**2**_-OH, noted as a); 4.01 (br s, C**H**_**2**_-O, *THOXMA*, noted as d); 4.85 (br s, -O**H**, noted as j); 7.4–7.9 (m, aromatic protons of CPDB, noted as i, k and m).

The conversions of co-monomers were calculated by ^1^H NMR (Equation (1)), via the comparison of signal integrals of the CPDB (7.4–7.9 ppm) and of the protons of -C=C- double bond of *THOXMA* (6.06–5.68 ppm) and/or *HEMA* (6.02–5.74 ppm).
(1)$$\mathrm{Conversion}\;(\%)=\frac{\left(I_{0,\;vinyl\;function\;}/I_{0,\;CTA}\right)-\left(I_{t,\;vinyl\;function\;}/I_{t,\;CTA}\right)}{\left(I_{0,\;vinyl\;function\;}/I_{0,\;CTA}\right)}\times 100$$
where $I_{0,\;CTA}$ and $I_{t,\;CTA}$ are the values of the integrals of the signal of the aromatic protons of the chain transfer agent (between 7.4 ppm and 7.9 ppm) at t = 0 and t, respectively; $I_{0,\;vinyl\;function}$ and $I_{t,\;vinyl\;function}$ are the values of the integral of the signal of one of the protons of the vinyl group of methacrylate (5.68 ppm and 6.06 ppm for *THOXMA*, 5.74 ppm and 6.02 ppm for *HEMA*) at t = 0 and t, respectively.

The DPs of the polymers were determined by ^1^H-NMR (Equation (2)), by comparing the integrals of the signals from the polymethacrylate backbone (*δ* = 0.51–2.1 ppm) with those of the CPDB (*δ* = 7.4–7.9 ppm).
(2)DP=Ipolymethacrylate backbone5ICTA5
where $I_{polymethacrylate\;backbone}$ is the value of the integral of the signal of the polymethacrylate backbone (between 0.51 ppm and 2.1 ppm), and $I_{CTA}$ is the value of the integral of the signal of the aromatic protons of the chain transfer agent (between 7.4 ppm and 7.9 ppm).

Molecular weight by ^1^H-NMR (M_n_) was calculated by Equation (3):(3)$$M_{n\;}=\left(DP\times Conv.\;HEMA\times M_{n,HEMA\;}\right)+\left(DP\times Conv.\;THOXMA\times M_{n,THOXMA\;}\right)+M_{n,chain\;transfer\;agent\;}$$
where $M_{n,chain\;transfer\;agent}$ = 221.34 g/mol, $M_{n,HEMA}$ = 130.14 g/mol, $M_{n,THOXMA}$ = 231 g/mol. DP (the degree of polymerisation) was calculated by ^1^H-NMR according to Equation (2). The conversion of *HEMA* and *THOXMA* co-monomers (Conv. *HEMA*, Conv. *THOXMA*) was calculated by ^1^H-NMR according to Equation (1).

### 2.7. Determination of the pK_a_ of the Monomers and Polymers

The pK_a_ of the monomers and corresponding polymers were evaluated by titration with a solution of HCl. Aqueous solutions of *THOXMA* (0.1 M) and *PTHOXMA* (0.1 M) were titrated with a solution of HCl 0.1 M. Each measurement of the pH (at different volumes of HCl added) was carried out in triplicate. The pK_a_ values were derived from the values of pK_b_ which were measured at the midpoint of the titration curves. The pK_a_ was calculated according to Equation (4).
pK_a_ = 14 − pK_b_
(4)

Additional experiments were performed in an NaCl aqueous solution (0.9 wt%) in order to evaluate the acido-basic properties of PTHOXMA in conditions similar with those of physiological medium.

### 2.8. Evaluation of the LCST of P(THOXMA_100_) and P(THOXMA_n_-stat-HEMA_m_) Statistical Copolymers

DLS measurements were performed at different temperatures ranging between 10 °C and 60 °C. The temperature corresponding to a sharp increase in the particle size was associated with the LCST of the statistical *P(THOXMA_n_-stat-HEMA_m_)* copolymers. The LCST of *P(THOXMA)_100_* was determined by extrapolation (to % HEMA = 0) on the plot of the evolution of the LCST value as a function of m. For each measurement, solutions of 1 g/L of the corresponding copolymers in three distinct buffers (pH = 4, 7.4 and 10) were analysed by DLS. The measurements were carried out in triplicate.

### 2.9. Cytotoxicity Assays

Cytotoxicity studies were performed using the mouse fibroblast cell line L929 (400620, CLS), as recommended by ISO10993-5. L929 cells were routinely cultured in Dulbecco’s Modified Eagle Medium with 2 mM L-glutamine supplemented with 10% fetal calf serum, 100 U/mL penicillin and 100 μg/mL streptomycin at 37 °C under a humidified 5% (*v/v*) CO_2_ atmosphere. In detail, cells were seeded at 10^3^ cells/mL (10^4^ cells per well) in a 96-well plate and incubated for 24 h. No cells were seeded in the outer wells. The medium was changed to fresh cell culture medium 1 h prior to treatment. Afterward, the cold polymer solution in 20 mM HEPES (4-(2-hydroxyethyl)-1-piperazineethanesulfonic acid) was added to the cells at the indicated concentrations (from 5 to 700 μg/mL), and the plates were incubated for 24 h. The control cells were incubated with fresh culture medium containing the same amount of HEPES as the treated cells. Subsequently, the medium was replaced by a mixture of a fresh culture medium and the resazurin-based solution PrestoBlue (prepared according to the manufacturer’s instructions). After further incubation for 45 min at 37 °C under a humidified 5% (*v/v*) CO_2_ atmosphere, the fluorescence was measured at λ_ex_ = 560 nm/λ_em_ = 590 nm with gain set to optimal, with untreated cells on the same well plate serving as negative controls. The negative control was standardised as 0% of metabolism inhibition and referred to as 100% viability. Cell viability below 70% was considered to be indicative of cytotoxicity. Experiments were conducted in six technical replicates. All experiments were conducted including blanks and negative controls.

### 2.10. Haemolysis Tests

The membrane damaging properties of the polymer were quantified by analysing the release of haemoglobin from erythrocytes. Sheep blood was provided by the Institute for Experimental Animal Science and Animal Welfare, Jena University Hospital. Briefly, sheep blood was centrifuged at 4500× *g* for 5 min. The pellet was washed three times with PBS (pH 7.4) by centrifugation at 4500× *g* for 5 min. Erythrocytes were suspended in PBS at pH 7.4 to resemble physiological conditions in blood/cytoplasm or in PBS at pH 6 to mimic the slightly acidic environment in the early endosome. The polymer was dissolved in cold 20 mM HEPES (4-(2-hydroxyethyl)-1-piperazineethanesulfonic acid) at a concentration of 7 mg/mL and diluted to 1 mg/mL. Cold polymer solutions of different concentrations in PBS with the respective pH were mixed 1:1 with cold erythrocyte suspensions and were incubated at 37 °C for 1 h. Erythrocyte suspensions were centrifuged at 2400× *g* for 5 min. The release of haemoglobin in the supernatant was determined at 544 nm. The absorbance was measured using a plate reader. Concurrently, determinations were conducted with washed erythrocytes either lysed with 1% Triton X-100 or suspended in PBS at the respective pH as a reference. The haemolytic activity of the polymer was calculated as follows (Equation (5)):(5)$$Hemolysis\;\;\%=\frac{A_{\left(\mathrm{sample}\right)}-A_{\left(\mathrm{PBS}\right)}}{A_{\left(\mathrm{Triton}\;X-100\right)}}$$

Here, A_(sample)_, A_(PBS)_ and A_(Triton X-100)_ are the absorbance of erythrocytes incubated with a respective sample, suspended in PBS and erythrocytes lysed with Triton X-100, respectively. The analysis was repeated with blood from three different animals.

### 2.11. Erythrocyte Aggregation

To investigate the behaviour of the pH-sensitive polymer towards cellular membranes at different pH values, red blood cells were treated with the polymer under physiological conditions in human blood (pH 7.4) and in a slightly acidic environment representing the pH of the early endosome (pH 6). Erythrocyte suspensions in PBS at different pH values were prepared and mixed 1:1 with polymer solutions as described above. After incubation at 37 °C for 2 h, erythrocyte aggregation was measured at 645 nm. As positive and negative assay controls, erythrocytes were treated with 50 µg/mL 25 kDa branched poly(ethylene imine) (bPEI) solution or PBS buffer at a respective pH. The aggregation activity of the polymer at different concentrations is given as an aggregation rate calculated as follows (Equation (6)):(6)$$Aggregation\;rate=\frac{1}{A_{\left(\mathrm{sample}\right)}}$$

Here, A_(sample)_ is the mean absorbance of a given sample.

## 3. Results and Discussion

### 3.1. Synthesis of Monomers and Hydrophobic/Hydrophilic Polymers

Two new methacrylates derived from thiomorpholine (named *THMA*) or thiomorpholine-oxide (named *THOXMA*) were prepared as presented in Figure 1. *THMA* was prepared by nucleophilic substitution of 2-bromoethylmethacrylate with thiomorpholine in the presence of K_2_CO_3_. The one-step reaction afforded *THMA* with a yield of 75% (Figure 1). The mild oxidation of THMA with H_2_O_2_ (aq., 30%) produced THOXMA with a yield of 80% (Figure 1). Monomers were obtained in sufficient purity without a need for elaborate purification methods such as distillation or column chromatography. ^1^H-NMR analysis confirmed the formation of both pure monomers (Appendix A). FTIR spectroscopy confirmed the presence of sulfoxide group (S=O) in THOXMA and the success of the oxidation reaction (Appendix A).

Since *THOXMA* is hydrophilic, one of the goals of this work was to study the homo- and co-polymerisation of this monomer, in order to develop new hydrophilic polymers. The RAFT process was used to prepare *P(THOXMA)_100_* homopolymer (Figure 2), as well as a series of *P(THOXMA_n_-stat-HEMA_m_)* statistical copolymers (Figure 3). All polymerisations were performed in DMSO using CPDB as chain transfer agent, while the pH was maintained at 4 to prevent CPDB degradation via hydrolysis. The molar ratios of monomer(s):CPDB:AIBN was kept at 100:1:0.25 in all polymerisations (Appendix A).

The ^1^H NMR spectrum of *P(THOXMA)_100_*, a new hydrophilic homopolymer, is shown in Figure 2. High conversion (i.e., 99.3%, by ^1^H-NMR) of *THOXMA* monomer was achieved after 6 h (Table 1, Appendix A). The linear increase in M_n_ over time (Figure 2B.), the linear first-order kinetic plot (Appendix A) as well as the low dispersity (~1.2) indicate that the homopolymerisation was well controlled.

*THOXMA* was then copolymerised with *HEMA* by RAFT. The structure and the characterisation results of all the (co)polymers are presented in Figure 3 and Table 1, respectively. Overall, the DPs of the resulting polymers (evaluated by ^1^H NMR, Appendix A) were above 80 and were close to the targeted DPs. High conversions (≥95% after 4 h, Table 1, Appendix A), determined by ^1^H NMR, were attained for all (co)polymerisations. Again, the linear evolution of ln(M_0_/M) with time (Appendix A), as well as the low dispersity (remaining relatively constant around 1.2) (Table 1) suggested that the polymerisations were controlled.

### 3.2. Acido-Basic Properties of THOXMA and PTHOXMA

The acido-basic properties of the monomer and polymer are summarised in Appendix A and shown in Appendix A. *THOXMA* displayed a pK_a_ value around 5.42 (Appendix A). This value corresponds to a slightly acidic character. The pK_a_ of *PTHOXMA* is around 5.57 (Appendix A). However, this value is lower than that of polymers presenting a tertiary amine function, such as poly(2-(dimethyl amino)ethyl methacrylate *PDMAEMA* (pK_a_ = 7.5) [22] and could be due to the steric hindrance of the heterocycle which renders the tertiary amine less accessible to protonation by the acid [23]. Hausig et al. [13] reported low pK_a_ values (between 6 and 7.1) for polymers comporting *N*-alkyl-piperazine units. Low pK_a_ values were also observed for poly(2-methyl-acrylic acid 2-[(2-(dimethylamino)-ethyl)-methyl-amino]-ethyl ester), [24] a polymer used in cellular transfection. The tertiary amine situated in the beta position of the ester function presented a pK_a_ around 5 [24]. In addition, low pK_a_ values (around 4.9) were reported by Butun et al. for poly(morpholine ethyl methacrylate) (*PMEMA*) [18]. According to their study [18], these unexpectedly low values of pK_a_ were a consequence of intra-molecular cyclisation (between the amino group of cyclic morpholine and the carbonyl groups of side chains) which decreased the overall basicity of the polymers. The same ring structure likely also explains the low pK_a_ of *PTHOXMA*. The presence, in *THOXMA*, of a sulfonyl group which is less electron withdrawing than an oxygen atom may explain the higher pK_a_ of *PTHOXMA* compared to *PMEMA*.

When the acido-basic properties of *PTHOXMA* were tested in a solution of NaCl (0.9 wt%), at a concentration specific to physiological serum, the pK_a_ values shifted slightly from 5.57 to 5.65 (Appendix A). This slight increase in the pK_a_ values in NaCl solution is related to the shielding effect of the salt that minimised the charge repulsion between the protonated amino groups. This result was consistent with previous works reported by Douglas et al. [25]. It is important to underline that the pK_a_ value of *PTHOXMA* around 5.6 is interesting for a potential use in biological applications such as cellular transfection [26].

### 3.3. Determination of LCST

LCST was determined using DLS. The cloud point for *PTHOXMA* was barely visible as the upper temperature limit was 60 °C. Therefore, the cloud point temperature of the copolymers with variable composition was used and the LCST was estimated via extrapolation to pure *PTHOXMA*. *HEMA* was chosen because it provides a structural ethyl-methacrylate pattern similar to *THOXMA* and because it is hydrophilic and biocompatible. The statistical *P(THOXMA_n_-stat-HEMA_m_)* copolymers were dissolved in a range of aqueous buffers (at pH 4, 7.4 and 10) and the particle size evolution vs. temperature was investigated at each pH, in order to determine the LCST of the various copolymers.

The particle size variation over temperature at physiological pH (pH 7.4) is presented in Figure 4A. All copolymers showed a sharp increase in the particle size in the temperature range examined (10–60 °C), which was considered as the cloud point temperature. As expected, the decrease in the *THOXMA* molar fraction (from 75% to 20%) shifted down the cloud point temperature from 56 °C to 42 °C. The decrease in the cloud point temperature could be explained by the intermolecular H-bonds induced by *HEMA*.

At pH 10, the polymers had a behaviour close to that of pH 7.4 (Figure 4B). The polymer cloud point varied from 52 °C (for a content of 75% in *THOXMA*) to 36 °C (for 20% *THOXMA* content).

Then, by extrapolation of the cloud point temperatures measured at different contents in HEMA to 0% HEMA, the apparent cloud point temperature of *PTHOXMA* was determined in both physiological and alkaline environments (Figure 5). At pH 7.4, the cloud point temperature was 65.4 °C, and at pH 10 the cloud point temperature was 57.9 °C.

In comparison, PMEMA of similar DP (around 100) and at a similar concentration (1% *w*/*v*) displayed cloud points around 36 °C at pH 7 and at lower temperatures at higher pH (8 and 10) [18]. The differences between PTHOXMA and PMEMA are thus subtle. PTHOXMA is more water soluble and shows higher cloud points than PMEMA at different pHs. This is likely a consequence of the slightly higher pK_a_ of PTHOXMA.

In contrast, at pH = 4, no cloud point was observed for PTHOXMA (or for PMEMA). In acid medium the particle size was constant (around 7 nm) over the complete range of temperatures scanned (10–60 °C) and for all copolymer compositions (Appendix A). This result is a consequence of the complete protonation of polymer chains at such acidic pH, leading to hydrosoluble polymers which were not sensitive to temperature.

### 3.4. Biocompatibility of PTHOXMA

In biological applications such as cellular transfection, the polymer that complex the genetic material must survive to the endosomal passage (where the pH is ~5–6) and thus prevent the degradation of the genes. Cationic polymers are currently used for this purpose, but they are toxic and damage the cell surface [19]. Thus, polymers acting as a proton-sponge system are highly investigated to overcome the endosomal barrier. Proton-sponge polymers are neutral at physiological pH (so before entering the endosome) but become charged in an acid environment (specific to the endosome). As it was presented before, *PTHOXMA* is neutral at physiological conditions, but become charged at a pH around 5.6, so it could emphasise an emerging potential for this application. Herein, the biocompatibility of *PTHOXMA* was assessed, by evaluating in vitro the cytotoxicity and hemocompatibility of this polymer. Copolymers of *THOXMA* and *HEMA* were not assessed since *PHEMA* is known to be biocompatible [27]. If both *PHEMA* and *PTHOXMA* are biocompatible, copolymers of *HEMA* and *THOXMA* will very likely be biocompatible too.

Firstly, the cytotoxicity of the *PTHOXMA* was studied on the mouse fibroblast cell line L929 using PrestoBlue assay, at pH 7.4 (Figure 6A). This assay works as a cell health indicator which uses the reducing ability of living cells in order to measure the cellular viability [28]. A cellular viability below 0.7 (or 70%) indicates cytotoxic behaviour [29]. These assays showed that *PTHOXMA* presented no cytotoxicity on L929 cells for concentrations below 400 μg/mL, while the cellular viability was above 0.9 (90%). These promising results further proved the interest to investigate *PTHOXMA* for blood compatibility.

Then, further investigations were conducted to study the red blood cell aggregation activity of *PTHOXMA* in sheep blood (i.e., pH 7.4) and in a slightly acidic environment representing the pH of the early endosome (i.e., pH 6). The results are summarised in Figure 6B, where the aggregation activity was expressed by the aggregation rate and compared to a polycationic commercial polymer (i.e., polyethyleneimine PEI) which is known for its high aggregation rate [30]. For both pH values tested and all *PTHOXMA* concentrations, an aggregation rate equal to 1 was observed. These results confirm that *PTHOXMA* did not cause the undesired cell aggregation.

Since *PTHOXMA* did not provoke the aggregation of red blood cells, further attempts were developed in order to investigate if the thiomorpholine oxide-containing homopolymer could damage the red blood cell membrane. To this regard, the release of haemoglobin from the erythrocytes was measured [29]. As evidenced in Figure 6C, the haemoglobin release was studied in two selected media: at pH 7.4 (to mimic the physiological conditions) and at pH 6 (specific for endosomal escape process which is a reference step in cellular transfection applications). At physiological pH (i.e., pH 7.4), haemoglobin was released in low amounts (below 1%), without significant influence of the polymer concentration in the blood medium. Haemoglobin release value below 2% is correlated with a non-haemolytic activity, [31] this result thus suggested that the *PTHOXMA* is blood compatible at pH 7.4 for concentrations between 10–100 µg/mL and that it did not damage the plasma membrane of the erythrocytes. In slightly acidic conditions (i.e., pH 6), a concentration-dependent haemoglobin release profile was observed. Particularly, an increase in *PTHOXMA* concentration from 10 µg/mL to 100 µg/mL led to an increase in the haemoglobin percentage from 0.8% to 6%, which indicates a variable haemolytic activity. For example, at the middle concentration of 50 µg/mL, a haemoglobin release slightly above 2% was detected. At this concentration, *PTHOXMA* was slightly haemolytic, so the blood cells were not significantly damaged. On the contrary, at 100 µg/mL, the *PTHOXMA* is highly haemolytic and it interacts with the cell membrane. Compared to the results obtained at physiological pH, the different hemocompatibility observed at pH 6 is likely a result of partial protonation of the thiomorpholine oxide heterocycles, since this pH is close to the pK_a_ of *PTHOXMA* (around 5.6). Despite this slight haemolytic activity at high concentrations, the results at pH 6 are very promising for a prospective use of *PTHOXMA* in cellular transfection application, which requires a polymer that promotes the release of active substances from the endosome. Statistical and block copolymers of *DMAEMA* and *MEMA* were also shown to be biocompatible and showed interesting results as transfection agents [17,32]. Given its in vitro non-cytotoxicity and hemocompatibility, *PTHOXMA* is also potentially suitable for drug delivery and transfection strategies. Its higher pK_a_ compared to *PMEMA* could also be advantageous for drug release approaches. These applications are under investigation in our laboratories and will be reported in due course.

## 4. Conclusions

New thiomorpholine oxide-containing polymers with tailored LCST at different pHs were developed in order to design stimuli-responsive materials for biological applications. Hydrophilic poly(thiomorpholine oxide ethyl methacrylate) polymers possessed a pK_a_ around 5.6 which denotes a weak acid character which could be exploited in biological applications such as cellular transfection. Interesting results were obtained concerning the behaviour of hydrosoluble *PTHOXMA* in environments with different pHs. At physiological and alkaline pHs, LCSTs around 65 °C and 58 °C were reported, while in acid conditions the polymer remained hydrophilic. In acid medium, the protonation of the *PTHOXMA* amino groups increased the solubility of the polymer and no aggregation occurred at any temperature. The high LCST values (65 °C in physiological conditions and 58 °C in alkaline conditions) open the gates to explore a large area of applications, such as thermal therapy in biomedicine (or thermal tumour ablation which requires temperatures above 50 °C) [33]. Lastly, *PTHOXMA* showed no-cytotoxicity and no haemolytic behaviour, without any cellular aggregation, which proved promising biocompatibility. To conclude, this study highlights the development of non-cytotoxic, blood compatible, pH- and temperature-responsive polymers based on thiomorpholine oxide ethyl methacrylate with tailored LCST, which may find applications in biosciences.

## Figures and Tables

**Figure 1 molecules-27-04233-f001:**
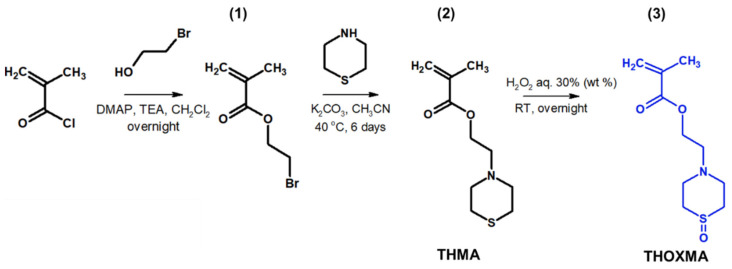
Synthesis of 2-bromoethylmethacrylate intermediary product (**1**), the synthesis of ethyl thiomorpholine methacrylate monomer THMA (**2**) and its oxidation to ethyl thiomorpholine oxide methacrylate monomer THOXMA (**3**).

**Figure 2 molecules-27-04233-f002:**
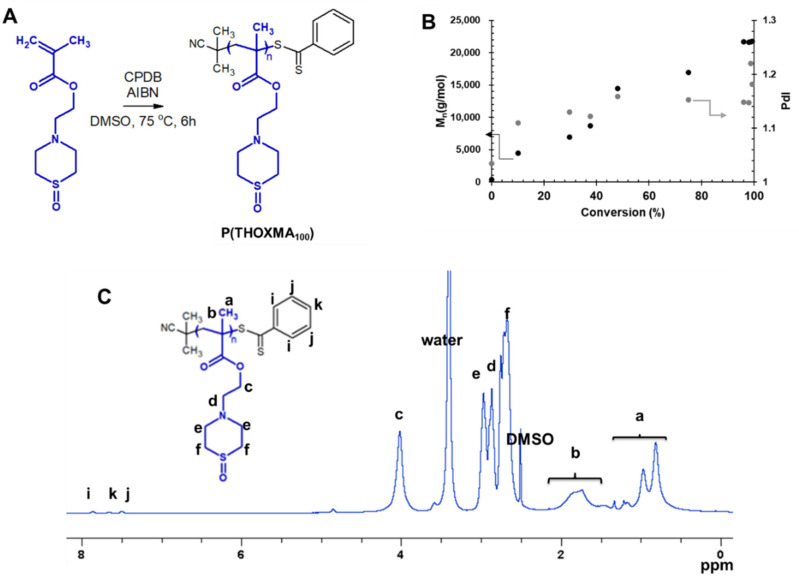
Synthesis of P(THOXMA_100_) (**A**); evolution of M_n_ and dispersity with conversion during the polymerisation of THOXMA (entry 1, Table 1) evaluated by SEC (**B**); ^1^H-NMR spectrum of P(THOXMA_100_) in DMSO d6 (**C**).

**Figure 3 molecules-27-04233-f003:**
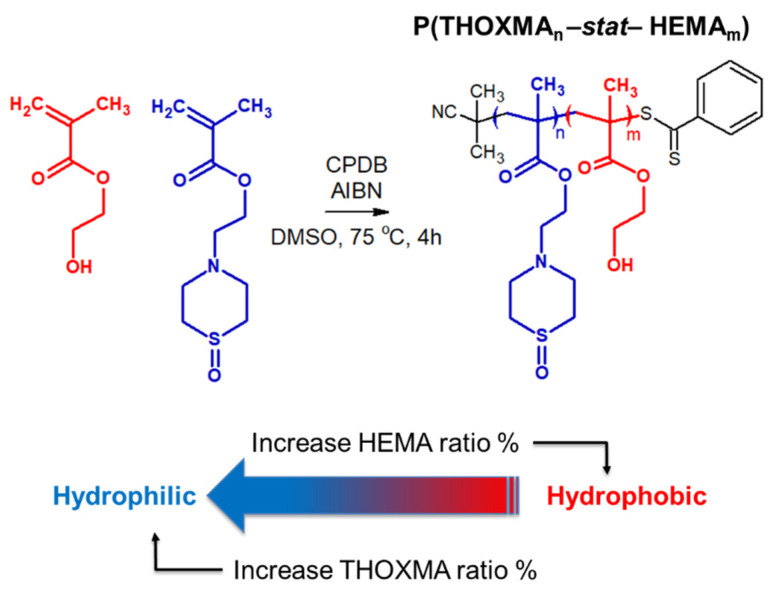
Structure of statistical copolymers of HEMA and THOXMA.

**Figure 4 molecules-27-04233-f004:**
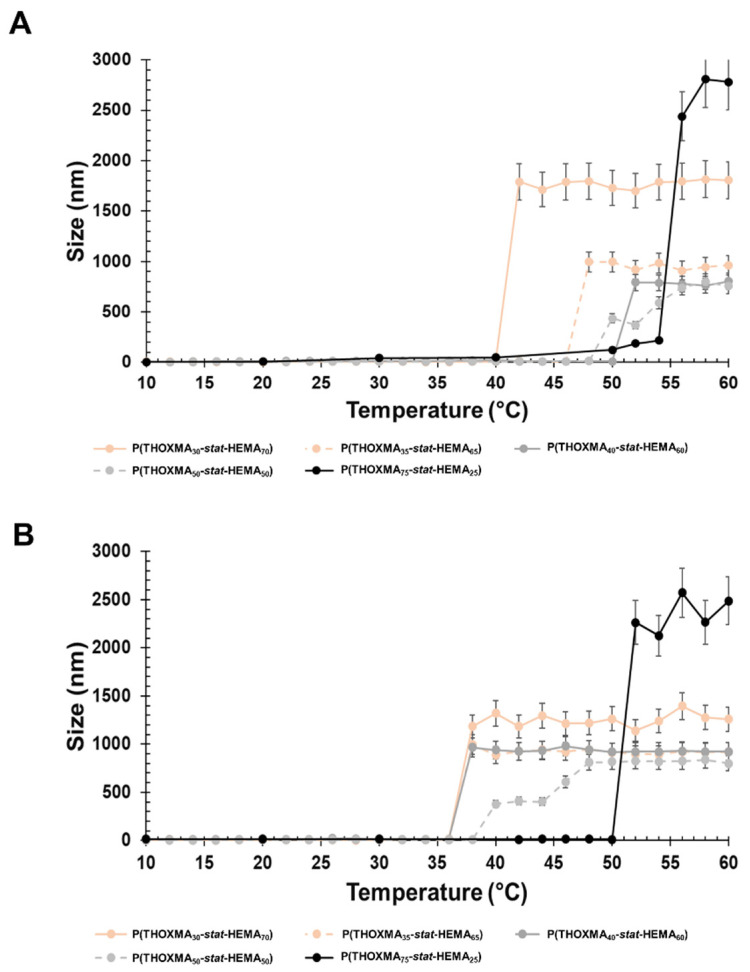
Particle size variation with the temperature of copolymer formulations (1 g/L) at physiological pH (pH 7.4) (**A**) and alkaline pH (pH 10) (**B**).

**Figure 5 molecules-27-04233-f005:**
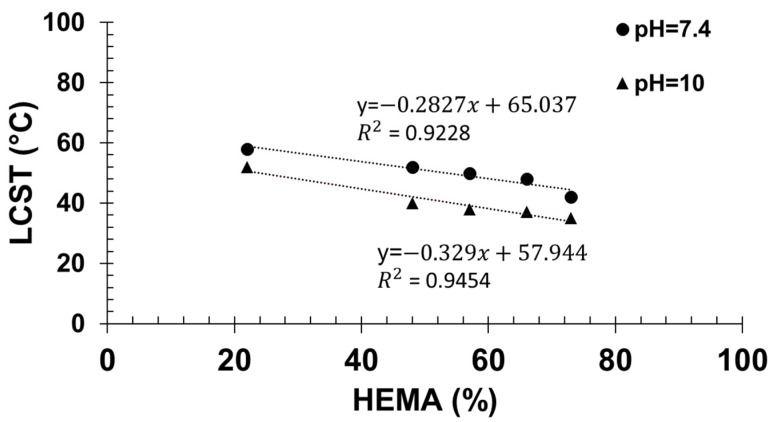
Determination of the LCST of PTHOXMA at pH = 7.4 and pH = 10.

**Figure 6 molecules-27-04233-f006:**
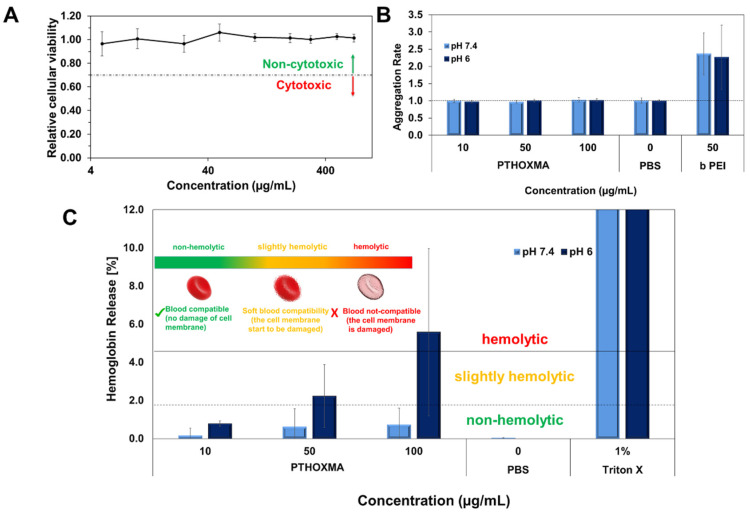
(**A**): Cellular viability results of *PTHOXMA* on fibroblast cell line L929 (at pH 7.4); (**B**): Aggregation activity of *PTHOXMA*; (**C**): Haemolytic activity of *PTHOXMA* evaluated by the release of haemoglobin from sheep blood erythrocytes.

**Table 1 molecules-27-04233-t001:** Characterisation data of homopolymers and statistical copolymers prepared by RAFT.

Entry	Conversion of Monomers (%) ^a^		Experimental DP ^b^	DP_Target_ ^c^	M_n_ (g/mol) ^d^	M_n_ (g/mol) ^e^	Dispersity (Đ) ^e^
	THOXMA	HEMA					
P(THOXMA_100_)	99.3	-	111	100	25,640	21,570	1.21
P(THOXMA_80_-*stat*-HEMA_20_)	95.6	98.1	80	100	16,900	14,100	1.19
P(THOXMA_50_-*stat*-HEMA_50_)	98	99	82	100	14,970	18,200	1.24
P(THOXMA_40_-*stat*-HEMA_60_)	97	99.5	81	100	14,600	16,100	1.22
P(THOXMA_35_-*stat*-HEMA_65_)	98.5	99.3	80	100	13,000	17,900	1.19
P(THOXMA_30_-*stat*-HEMA_70_)	99	99.2	82	100	13,100	19,110	1.2

(a) Calculated by ^1^H NMR using Equation (1). (b) Calculated by ^1^H NMR using Equation (2). (^c^) Calculated using the following equation DPtarget = (([HEMA]/[CPDB]) × Conv HEMA) + (([THOXMA]/[CPDB]) × Conv THOXMA). (d) Calculated by ^1^H NMR using Equation (3). (e) SEC analysis performed in DMF containing 0.1% LiCl and by using PMMA standards.

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
