# Peer review of "Stimuli-Responsive Thiomorpholine Oxide-Derived Polymers with Tailored Hydrophilicity and Hemocompatible Properties"

_molecules, 2022, doi:10.3390/molecules27134233_

Round 1

Reviewer 1 Report

      This paper describes the synthesis and properties of poly(ethylthiomorpholine oxide methacrylate) (PTHOXMA) and its copolymers comprising the other repeating units derived from hydroxyethyl methacrylate.  The structural motif of PTHOXMA is poly(morpholine ethyl methacrylate) (PMEMA).  The syntheses of PTHOXMA and its copolymers are performed via reversible addition-fragmentation chain-transfer (RAFT) polymerization.  The pKa value of PTHOXMA is determined to be 5.57, which appears to be slightly higher than that of PMEMA (pKa 4.9).  The LCST behaviors of PTHOXMA are evaluated by DLS using PTHOXMA solutions (1 g/L) in pH 4.0, 7.0, and 10.0 aqueous media.  The biocompatibility of PTHOXMA is discussed through the results of cytotoxicity assays, hemolysis tests, and erythrocyte aggregation.

      The main problem of this manuscript is that the authors fail to provide the comparable data of PTHOXMA to those of PMEMA.  While the structure of PTHOXMA is very similar to PMEMA, the preparation of PTHOXMA appears to be more costive and labor-intensive rather than that of PMEMA.  The synthetic approach itself of PTHOXMA is not so new.  Therefore, the LCST behaviors and the biocompatibility of PTHOXMA should be highlighted by the comparison with those of PMEMA, considering the demand as the article of molecules. 

      When the authors will investigate the comparable experiments, I think that the effects of concentration of PTHOXMA on the DLS results should be evaluated.

      At all, this paper is not acceptable at the present form, calling for major revision.

Author Response

We thank the reviewer for the attentions he or she paid to our manuscript and for his/her comments. Please find our point by point answer below.

 The main problem of this manuscript is that the authors fail to provide the comparable data of PTHOXMA to those of PMEMA.  While the structure of PTHOXMA is very similar to PMEMA, the preparation of PTHOXMA appears to be more costive and labor-intensive rather than that of PMEMA.  The synthetic approach itself of PTHOXMA is not so new.  Therefore, the LCST behaviors and the biocompatibility of PTHOXMA should be highlighted by the comparison with those of PMEMA, considering the demand as the article of molecules

Response: We thank the reviewer for this constructive comment. The article was modified to better compare the solution behaviour of PMEMA and PTHOXMA. PTHOXMA possesses higher LCST than PMEMA at similar molar masses and concentration. PTHOXMA is also slightly less acidic than PMEMA. Statistical and block copolymers of DMAEMA and MEMA were also shown to be biocompatible and were showed interesting results as transfection agents. Given its in-vitro non-cytotoxicity and hemocompatibility, PTHOXMA is also potentially suitable for drug delivery and transfection strategies. Its higher pKa compared to PMEMA could also be advantageous for drug release approaches. These applications are under investigation in our laboratories and will be reported in due course.

      When the authors will investigate the comparable experiments, I think that the effects of concentration of PTHOXMA on the DLS results should be evaluated.

Response: We respectfully disagree with this comment. The salient point of the present article was to demonstrate the suitability of PTHOXMA as a potential substitute for other biocompatible, non–cytotoxic polymers (such as PEG for example). The study of the effect of concentration on the LCST behavior of PTHOXMA is out of the scope of this article (as is the study of the molar mass dependency of these properties). The complete study of the solution properties of PTHOXMA will be the subject of another manuscript.

Reviewer 2 Report

The manuscript by Arsenie reported the preparation of non-cytotoxic, blood compatible, pH and temperature responsive polymers based on thiomorpholine oxide ethyl methacrylate with tuneable lower critical solution temperature. Generally, the work is interesting and well presented. I can recommend acceptance of it after following points are addressed.

1)  Table 2 is not necessary as the data summarized in it is very simple.

2)  In the y-axis of figure 4, the numbers for the size are hard to tell from current scales.

3)  It needs to explain why the authors only studied the biocompatibility of PTHOXMA, not the copolymers as well?

Author Response

We thank the reviewer for the attentions he or she paid to our manuscript and for his/her comments. Please find our point by point answer below.

  • Table 2 is not necessary as the data summarized in it is very simple.

Response: Table 2 was removed from the main manuscript, and added to the Supplementary Information file as Table S3.

  • In the y-axis of figure 4, the numbers for the size are hard to tell from current scales.

Response: In the initial manuscript, the scale of the y-axis was represented in logarithmic format. The y-axis is now represented in linear scale for clarity.

  • It needs to explain why the authors only studied the biocompatibility of PTHOXMA, not the copolymers as well?

Response: PHEMA has been reported to be biocompatible and non cytotoxic. Copolymers of HEMA and other biocompatible non cytotoxic comonomers have also been reported to be biocompatible and non cytotoxic. The emphasis of the present article is THOXMA and PTHOXMA, and the crucial question was whether PTHOXMA is biocompatible or cytotoxic. This is why the copolymers of THOXMA and HEMA were not tested. The following sentence was added to the manuscript.

“Copolymers of THOXMA and HEMA were not assessed since PHEMA is known to be biocompatible. If both PHEMA and PTHOXMA are biocompatible, copolymers of HEMA and THOXMA will very likley be biocompatible too.”

Round 2

Reviewer 1 Report

I accept the author's responses and the revisions to my comments.  Therefore, I judge that the present manuscript is acceptable for the publication in Molecules as an article.